

# A hybrid data assimilation method and its comparison with an Ensemble Optimal Interpolation scheme in conjunction with the numerical ocean model using altimetry data

**Konstantin Belyaev[1,2], Andrey Kuleshov[2], Ilya Smirnov[3] and Clemente A. S. Tanajura[4]**

[1]Shirshov Institute of Oceanology of Russian Academy of Sciences, Moscow, 117997, Russia
[2]Keldysh Institute of Applied Mathematics of Russian Academy of Sciences, Moscow, 125047, Russia
[3]Lomonosov Moscow State University, Faculty of Computational Mathematics and Cybernetics, Moscow, 119991, Russia
[4]Federal University of Bahia, Physics Institute and Center for Research in Geophysics and Geology, Salvador BA, 40170-280, Brazil

**Correspondence:** Andrey Kuleshov (andrew_kuleshov@mail.ru)

**Abstract.** An original hybrid data assimilation scheme recently developed is presented and tested. The scheme is based on the application of the theory of diffusion random processes. It is applied here in conjunction with the Hybrid-Coordinate Ocean Model (HYCOM) to assimilate altimetry data from the Archiving, Validating and Interpolating Satellite Oceanography Data (AVISO) in the Atlantic. Several numerical experiments were conducted and their results were analyzed. It is shown that the method is able to assimilate data and to produce analyses closer to observations. It also conserves the model balance. This method allows calculating the confidence range of the analyses by estimating their errors The presented method is compared with the Ensemble Optimal Interpolation scheme (EnOI) and it is shown that it has several advantages, in particular, it provides a better forecast and requires less computational cost.

## 1 Introduction

A data assimilation (DA) theory, as a part of mathematical and numerical research, is a scientific area of a great practical importance. The ideas and methods of DA are used in ocean modelling, weather forecast, operational oceanography and many other science fields. Its main goal is to combine the numerical model results with independently observed data in a mathematically optimal way to represent the physical system state. Since the two extreme approaches for this representation are obviously poor - namely, ignoring observations at all or replacing only the observed model variables by observational data without changing the others - the optimal way on how to produce the optimal representation of reality is not trivial.

It is important to point out that the terms "optimal", "combine" and so on have minor sense unless they are defined within a strong theoretical framework. Therefore, in order to give meaning to these formulations, it is necessary to consider the DA problem as a part of one or several mathematical and/or physical theories, such as the optimal control theory, mathematical statistics, numerical analysis, and others.

Studies to tackle the representation of physical systems behavior have been carried out and published in the scientific literature for more than 50 years, since the beginning of the 1960s. A good review of the main achievements in this direction during the last century is presented in (Ghil et. al., 1991). From the beginning of the 2000s, the main progress is related to the development of computer facilities, explosion in observational data network, parallel computations and other technical novelties. This advance leads to the progress in new mathematical methods and algorithms, the construction and development of numerical models with very high resolution, the data exchange all over the world, etc. At the present time, the DA techniques, algorithms and methods have become the essential part of operational oceanography on the ocean shelf and coastal zones, especially in the oil and gas mining zones, as well as in the zones of pipeline transportation. Several national and international scientific projects are specially aimed to seek the optimal solution of the DA techniques in conjunction with the





regionally configured numerical models. In particular, one can indicate the Brazilian REMO project (Lima et. al., 2016), Australian Blue Link project (Schiller et. al., 2011), American HYCOM&NCODA project (Cummings et. al., 2013) and others.

However, the development and application of new and more advanced DA schemes and methods remains an actual and very important theoretical and practical task. There is a necessity to have a powerful and, at the same time, relatively portable and economy DA scheme which would be applicable to various numerical ocean and coupled ocean-atmosphere models and would provide a satisfactory and reliable forecast of the ocean characteristics in short and media-term periods.

Many of used DA methods and approaches may be divided into two large groups. First group is based on the functional scheme; its modern version is known in literature as 4D_VAR approach. Its main idea is to find a model trajectory within given time-window such as this trajectory passes nearby the observations as close as possible with respect to the minimum of a known functional. Actually this approach reduces to the solution of an inverse problem in order to find initial conditions of the model and to obtain the model trajectory started from the sought fields close to observations with respect to the given metric of closeness. Normally, one should minimize some functional, as a function of a seeking trajectory in a known metric, which is given a priori. Several modern research in this direction can be found in (Marchuk et. al., 2012; Talagrand et. al., 1987; Agoshkov et. al., 2010).

The second large group is known as a "dynamical-stochastic" DA approach. This approach uses the theory of statistical estimation or filtration to find out the best estimator to minimize the variance of this sought field among all others theoretically possible fields. Usually, the observed variables are originally supposed to be a sum of a known signal, which represents the model and stochastic noise with the known probabilistic properties. Started from the so-called "objective interpolation method" (see for instance (Penduff et. al., 2002)), its modern development is known in the literature as the Ensemble Kalman filter method (EnKF). As the examples one may refer to the papers (Evensen, 2009; Xie et. al., 2010).

There are also, relatively small numbers of papers where the considered DA methods distinguish from that two groups. For instance, in (Van Leeuwen, 2015) the corresponding scheme based on the partial method is applied. Also, in (Van Leeuwen, 2011) the well-known Bayes technique is used to find the best estimation of a posteriori probability after one time-step model simulation. However, the most actually used DA techniques relate to the schemes indicated before.

Recently the hybrid DA schemes combining the both approaches are appeared. In those schemes both ideas, namely, functional and dynamical-stochastic are exploited together. In particular, the minimum of functional is sought in the metric which represents the variance of the stochastic variable or variables that are determined from the observations. Some examples may be found in (Lorenc et. al., 2015; Tanajura et. al., 2013). We may also refer to the work (Tanajura et. al., 2009), where the hybrid DA method based on theory of diffusion stochastic processes was proposed.

This study utilizes the ideas published in (Tanajura et. al., 2009) but deals with the application of the novel DA method created in (Belyaev et. al., 2018). The method hereafter will be referred to as GKF, Generalized Kalman Filter. Unlike the standard EnKF, which takes into account at the assimilation moment only the model output called background and observed data, the GKF accounts additionally the temporal tendency both from the model and observations. The explicit form of the obtained gain matrix (analogous of the Kalman gain matrix in the standard scheme) contains the time-derivative of the model and of the ensemble average of observational fields. As a consequence, the gain matrix turns out to be zero if these tendencies coincide and, hence, no assimilation occurs. Therefore, the GKF becomes better in a sense of numerical forecast of the ocean state than its counterpart, EnKF. The mathematical derivation of the GKF is not trivial, and the paper (Belyaev et. al., 2018) specially is dedicated to its formulation and mathematical proofs, but its physical sense is rather transparent. The method is based on the undeniable physical axiom, namely, on the path-of-least-resistance principle. According to this principle, all transitions from one physical state to other pass with the minimum energy, which may be set by the specific Lagrange functional. Since actually, the assimilation process is the transform from the model ocean state without specific data to the model state including the new coming information, namely the information from data, the corresponding transition must take





place and this transition is realized with respect to this principle. Paper (Belyaev et. al., 2018) contains all the mathematical aspects of these ideas, but this study is focused on its application, feasible realizations and analysis of the results.

Here, this method is used in conjunction with the ocean model HYCOM, presented in (Bleck, 2002; Bleck et. al., 1981). It the current work along with its application to the dynamical simulation in the Atlantic, it also is compared with the standard
Ensemble Optimal Interpolation (EnOI) method (Evensen, 2009), a simplified version of the EnKF, as an alternative data assimilation scheme. Twin experiments have been conducted for the same initial conditions and with the assimilation of the equivalent data. The AVISO archive (www.aviso.org) was chosen as the input data for the assimilation.

Concisely, it is possible to point out how two or more DA methods may be compared to. There are several criteria of their skills including computational consumptions, feasibility of their realization, reliability of their implementations etc. However,
the most obvious criterion of comparison may be determined as the minimum of variance of the forecast error. Indeed, the basic idea to use any data assimilation technique is to minimize the model forecast error if the model starts from the corrected field after assimilation. If one DA method does it better then another, it is obviously preferable. This method of comparison has been used earlier in papers (Belyaev et. al., 2012), where the standard method of objective interpolation, OI has been compared with the EnKF and method, based on the application of the Fokker-Planck equation (Tanajura et. al., 2009).
Similarly, the OI method has been compared with the EnOI accordingly to this criterion in (Kaurkin et. al., 2018).

The main goals of the study are the following: (i) to present the feasibility and applicability of the referred GKF method including its parallelization and computational effectiveness; (ii) to compare this method with the alternative EnOI scheme and to prove that the GKF has many advantages including less computational consumptions and better forecast properties; (iii) to analyze the modelling results and to show that the GKF captures the basic structure of synoptic variability in the Atlantic;
(iv) to estimate the analysis error appeared in the model assimilation and its dynamics in time.

The structure of this work is the following: Section 1 is the introduction, Section 2 is the description of the DA method and of the algorithm of its realization, Section 3 is the analysis of the results and the comparison with the EnOI method and with the control, i.e. the model simulation from the same initial conditions and forcing but without assimilation. Section 4 contains the estimation of the model forecast error variance and its dynamics in time. Ultimately section 5 presents the conclusions and
prospects for further developments.

**2 The assimilation method and the numerical algorithm of its realization**

**2.1 The mathematical description of data assimilation scheme GKF**

Let the mathematical model be governed by the equations

$$\frac{\partial X}{\partial t} = \Lambda(X, t) \tag{1}$$

with the initial condition $X(0) = X_0$.

Hereafter, $X$ means the model state vector defined on a phase-space, i.e. on the set of values which model variables can take, $t$ is time, $\Lambda$ denotes the vector-function defined on the same phase-space and on a time-interval $[t_0, T]$. Without loss of generality $t_0$ is assumed to be 0. In discrete realization, the model state vector has a dimension $r$, where $r$ is the number of grid points multiplied by the number of independent model variables. On the time interval $[0, T]$ the discretization $0 = t_0 < t_1 <$
$\ldots < t_N = T$ is introduced. For simplicity and also without loss of generality all these moments are assumed to be equidistant $\Delta t = t_{n+1} - t_n$. On each time interval $[t_n, t_{n+1}]$, $n = 0, 1, \ldots, N-1$ model equations are numerically solved and at moment $t_{n+1}$ data assimilation is performed by the formulae (Belyaev et. al., 2018)

$$X_{a,n+1} = X_{b,n+1} + K_{n+1}(Y_{n+1} - H X_{b,n+1}), \tag{2}$$

$$K_{n+1} = (\sigma_{n+1}^2)^{-1}(\Lambda_{n+1} - C_{n+1})(H \Lambda_{n+1})^{\mathrm{T}} Q_{n+1}^{-1}, \tag{3}$$





$$\sigma_{n+1}^2 = (H\Lambda_{n+1})^{\mathrm{T}} Q_{n+1}^{-1}(H\Lambda_{n+1}), \tag{4}$$

where $X_{a,n}, X_{b,n}, \ n = 0,1,\ldots,N$ are the model state vectors before and after assimilation, respectively, i.e., the analysis and background; $Y_n$ is the observation vector at the same moment having a dimension $m$, where m is the number of observation points, multiplied by the number of independently observed variables. It is assumed that $X_{a,0} = X_{b,0} = X_0$ is the known initial

condition. Further, $K$ is the gain matrix (analogous to the Kalman gain matrix) with dimension $r \times m$, $H$ is the observational projection matrix with a dimension $m \times r$. This projection interpolates the observed variables from model grid points onto points of observations and simultaneously eliminates all redundant, i.e. not observed variables. As usual, the superscript T denotes the transpose of a vector and/or a matrix.

Two variables in Eqs. (2)–(4) are suggested to be known, namely $C_n$ which is the observational trend, defined by the

formula $C_{n+1} = \frac{E(\hat{Y}_{n+1} - X_{a,n})}{\Delta t}$, where $\hat{Y}_{n+1}$ means the extended observational vector, that is a vector which coincises with observations for observational phase-space but is prolonged over on entire model phase-space; symbol $E$, ordinary, stands for the mathematical expectation or ensemble average; $\Lambda_{n+1} = \frac{X_{b,n+1} - X_{a,n}}{\Delta t}$, and $Q_{n+1} = E(\tilde{Y}_{n+1} - HX_{a,n})(\tilde{Y}_{n+1} - HX_{a,n})^T$, where $\tilde{Y}_{n+1} = Y_{n+1} - HC_{n+1}$. Actually, $\tilde{Y}_{n+1}$ is the anomaly relatively the observational trend. According to definition $E(\tilde{Y}_{n+1} - HX_{a,n}) = 0$.

This scheme with all necessary and sufficient conditions was introduced in (Belyaev et. al., 2018). Also, it was shown that this scheme generalizes the standard Kalman scheme method which follows from Eqs. (2)–(4), if $C_n = 0$ and $X_{a,n}$ coincides with the ensemble average.

## 2.2 The numerical definition of known parameters

As is seen from Eqs. (2)–(4), this algorithm can be applied to arbitrary numerical model with any physically reasonable initially

conditions and it gives the output result (analysis) at any moment of time $t_n, n = 0,1,\ldots,N$. To provide its correct realization, it is necessary and sufficient to set up two aforementioned parameters, namely, the observational drift vector $C_n$ and the error covariant matrix $Q_n$. To set up the vector $C_n$, we apply the following algorithm: previously, using the Monte-Carlo method the ensemble statistics from $M$ independent model runs is created. Theoretically, this ensemble should statistically represent the unknown "truth" value of field $X$ which satisfies the Eq. (1). Let this ensemble at each of moment of assimilation be

denoted as $\hat{X}_n^j, n = 0,\ldots,N; j = 1,\ldots,M$. Let us, suppose also, that the analysis fields $X_{a,n}, n = 0,\ldots,k$, started from known initial condition $X_0$ until moment $k$ where $k<N$ are already constructed. Then, the observational trend $C_{k+1}$ for each grid point is found out according to formula

$$C_{n+1} = \left(M^{-1}\sum_{j=1}^{M}\hat{X}_{n+1}^j - X_{a,n}\right)/\Delta t. \tag{5}$$

For the covariance $Q_n$ one may act similarly. If the ensemble statistics for moment of assimiation $t_i$ is known then $Q_n$ is

calculated by formula

$$Q_{n+1} = M^{-1}\sum_{j=1}^{M}(H\hat{X}_{n+1}^j - HX_{a,n})(H\hat{X}_{n+1}^j - HX_{a,n})^{\mathrm{T}}. \tag{6}$$

It is important to point out that in Eq. (6) only the observed components of entire model state vector X are used. In the expanded form of Eq. (6) we consider the multiplication of the values at each pair of grid points but only between observed variables.

In (Belyaev et. al., 2018), it is proved that if the condition $E(\tilde{Y}_{n+1} - HX_{a,n}) = 0$ holds, than this construction of drift

vector $C_n$ and covariance matrix $Q_n$ really estimates the observational trend and error covariance matrix of observational anomalies. Finally, using the Birkhoff –Khinchin theorem (Kolmogorov, 1938), which states that the ensemble average is approximated by the temporal average for large enough numbers of series, it is possible to rewrite Eq. (6) as

$$C_{n+1} = (n+1)^{-1}\sum_{i=1}^{n+1} X_{a,i} - X_{a,n}, \tag{7}$$





where $i$ is the number of assimilation time-steps until assimilation moment $n+1$. Our experiments showed that for $n >10$ Eq. (7) provides a good approximation of Eq. (5) but requires much less number of computations.

## 3 Computational experiments

### 3.1 The model and observational data base

Several numerical experiments have been carried out with the AVISO data and HYCOM model. The model HYCOM which has been used only as a tool to perform the assimilation experiments is very well-known (Bleck, 2002; Bleck et. al., 1981) and there is no need to give its detailed description. The model has been configured as follows: its version 2.2.14 has a spatial resolution of approximately 0.25 degree in the Atlantic in both West-East (OX axis) with 420 grid points, and South-North (OY axis) with 720 grid points directions. This version considers 21 vertical layers of equal density from top to bottom

(OZ axis) The model domain covers the major part of the Atlantic from Antarctica and up to 55°N. It includes the Caribbean Sea and Mexican Bay but excludes the Mediterranean Sea. On the lateral boundaries and on the sea bottom the Dirichlet conditions, i.e. the fixed climatological values are set up.

The model computes 4 barotropic variables (sea level, two velocity components and barotropic pressure on the sea surface) and 105 baroclinic components, namely, 5 variables on each given density layer: temperature, salinity, two velocity

components, and layer thickness. After assimilation all of those variables change with respect to Eqs. (2)–(4). Therefore, totally the dimension of model state vector, denoted above as $r$ was 480x720x109.

For assimilation the AVISO archive of sea level anomalies (SLA) relatively the temporal average for 10 years (2002-2011yr) has been utilized. This archive has been downloaded and prepared for assimilation from website (www.aviso.org). Previously it was undergone the quality control, the details of this procedure are out of the scope of this paper. As the result of

the quality control a part of observational array has been excluded from consideration. However, about 10000 daily recorded data remain to be used in the assimilation experiments. Therefore, the dimension of vector Y denoted above as $m$ was about 10000 per day. Consequently, the size of the matrices used in Eqs. (2)–(4) has an order ~$10^9$ and must be reduced due to parallelization technique.

It is also reasonable to mention that the several studies related to the assimilation of AVISO data into HYCOM were

presented earlier, in particular, in (Tanajura et. al., 2015). There the used assimilation method is distinguished from the method in the current work. Below we compare the our GKF assimilation scheme and the used scheme before on the same data base and the similarly configured model.

### 3.2 Parallelization of computations

Since this DA method is new, it is necessary to briefly describe the program realization of the presented scheme. The

assimilation is realized as a separate program module DAM (Data assimilation module) on Fortran 95 with the usage of MPI (message passing interface) library. The calculated model variables are entered in DAM which works independently on the model blocks and utilizes the other processing decomposition of the model domain. It is reasonable to note that the size of 3D model arrays requires several GB. The high parallelization is based due to the independency of GKF scheme in each observational point, Eqs. (2)–(4). Also, it should be point out that there is a necessary to storage the large data volume in

operative computer memory with the memory limit about 1.5-2 GB on each node. To optimize the assimilation processes all MPI- nodes have been combined in several blocks for 8 elements in one block. This substantially accelerates the computational process and requires about 40 minute of computer elapsed time on 24 nodes instead the 4.5 hours for on node. The module DAM has been realized on HPC "Lomonosov 2" of the Lomonosov Moscow State University (Voevodin et. al., 2012).

### 3.3 The algorithm of modelling





The assimilation experiments have been performed as follows. Previously, the Spin Up run for the HYCOM has been executed for 40 years from the rest and forced with the NCEP climatological reanalysis (Kalnay et. al., 2002) of wind stress and heat fluxes. Then the last 10 years of the modelled output have been saved and daily recorded. That means that we have an archive with 10 completely defined fields of 109 model computed variables for each calendar day from January 1 until December 31
(Day February 29 was excluded from consideration).

The numerical experiments start from January 1 of 2010, forced by real wind stress and heat fluxes recorded on this day from GFS (global forecast system, NCEP) and lasts 1 month until January 31, 2010. Previously, the meteorological data were interpolated onto model grid. The assimilations are executed according to Eqs. (2)–(4) daily. Information from daily recorded model outputs is used further for estimation of observed trend $C_n$ and observed anomalies $Q_n$ at moment of assimilation n which
coincises with number of day in its chronological order, from 1 until 31.

The samples for the assimilation experiments according to Eqs. (5)–(7) are constructed as follows : the ensemble statistics for day *n* have been set from the given climatological model output with the same number and plus-minus four equidistanly obtained days with interval two , totally 50. Other words : to set up the ensemble statistics on day 15 we take the climatological model output on January 11,13.15.17,19. Therefore, the length of sample is 50.
Three type of the experiments have been carried out. The control experiment A0, where model is integrated for one month without any assimilation. A1– the assimilation experiment with EnOI method and A02 – the assimilation experiments with GKF method.

### 3.4 Comparison with standard scheme EnOI

For the comparison with the GKF, the standard EnOI scheme with the same data and the same model outputs used to create
the ensemble was applied. Once it is done, the analysis is computed according to the Eq. (2), where

$$K_{n+1} = \alpha B_{n+1} H^T (H B_{n+1} H^T + R)^{-1}, \qquad (8)$$

$$B_{n+1} = M^{-1} \sum_{j=1}^{M} (\hat{X}_{n+1}^j - \bar{X}_{n+1})(\hat{X}_{n+1}^j - \bar{X}_{n+1})^T, \qquad (9)$$

$\bar{X}_{n+1} = M^{-1} \sum_{j=1}^{M} \hat{X}_{n+1}^j$ means the ensemble average and for other notations we used the same ones as in Eqs. (2)–(4). The instrumental error covariant matrix $R$ and the empirical scalar α are defined "manually" from heuristic considerations.
Equations (8) and (9) are widely known and as it was shown earlier in [16] they follows from Eqs. (2)–(4) if $C_{n+1} = 0$ and $\bar{X}_{n+1} = X_{a,n}$.

### 4 Results of the experiments and their analysis

As observational data there are used the satellite ocean measurements from archive AVISO, where data are recorded along the satellite tracks (Fig. 1). As an average for one day there are recorded about 30 thousand values while the model
computes much more sea level output values depending on the grid resolution.  In order to make a correct comparison of the model output with observations it is necessary to project the model values onto observational points.  The observed values may be both greater or smaller their counterpart model values. Fig. 1 shows all values where the difference between observations and projected model values is positive by the red and by the blue it is shown the contrary.

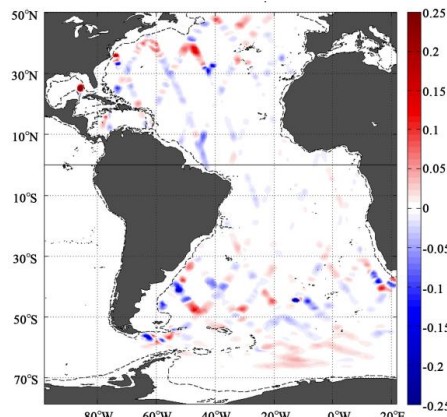

**Figure 1.** Model domain and satellite tracks over the Atlantic. Red points show where data value exceeds the model one; blue points show the opposite.

To compare the skills of different DA methods numerically we introduce the following values. Let

$$var_n = L^{-1} \sum_{i=1}^{L} ((SLA_m)_n^i - (SLA_o)_n^i)^2 \tag{10}$$

be a variance of the model error at time-moment $n$, that is the difference in squared between the model variable interpolated onto observational point $i$ , in particularly sea level anomaly, denoted as $(SLA_m)_n^i$ and corresponding observation value, denoted as $(SLA_o)_n^i$. This difference is taken over all observational points at each moment $n$ from $n=1,2,...,30$ with total

amount of observations equled $N$ . This amount $N$ depends on moment $n$ is different and it is not explicitly shown. Along with the variable $var_n$ we will consider two other variables $var_{f,n}$ and $var_{a,n}$ that are defined by formulae

$$var_{f,n} = L^{-1} \sum_{i=1}^{L} ((SLA_f)_n^i - (SLA_o)_n^i)^2, \tag{11}$$

$$var_{a,n} = L^{-1} \sum_{i=1}^{L} ((SLA_a)_n^i - (SLA_o)_n^i)^2, \tag{12}$$

where $(SLA_f)_n^i$ and $(SLA_a)_n^i$ are respectively the forecast and analysis model values at moment $n$ at observational point $i$. The

forecast value $(SLA_f)_n^i$ means the model forecast at moment $n$ if the assimilation has been done on the previous time-moment, $n-1$, but the analysis value $(SLA_a)_n^i$ is counted at the same time-moment.

The time-behavior of three mean root square variables $var_n$, $var_{f,n}$ and $var_{a,n}$ respectively is presented in Fig. 2 and Fig. 3. Figure 2 contains the analysis error for three DA methods including control but Fig. 3 shows the forecast errors for the same scheme. It is obvious, the control is unchanged in both Figs.

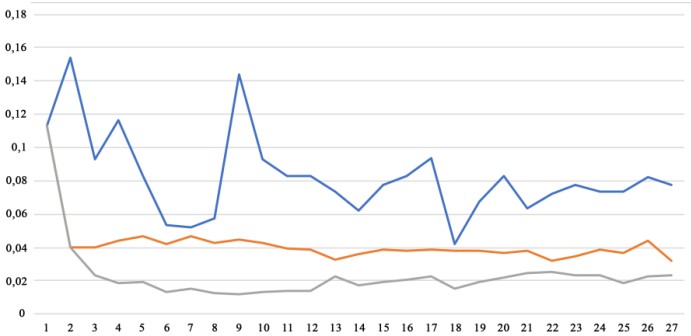


**Figure 2.** Analysis error variance for 3 model runs. Blue line is the model control error, yellow line is the EnOI model error, gray line is the GKF model error.

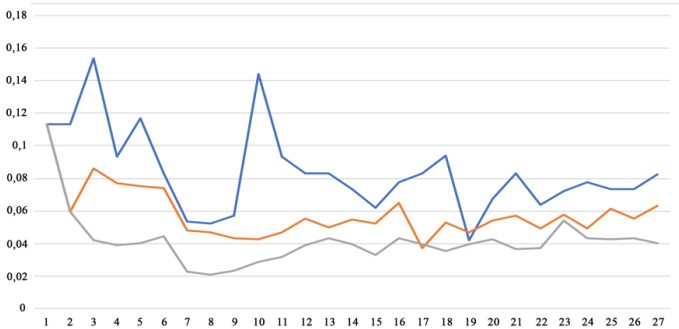

**Figure 3.** 24h forecast error variance for 3 model runs. Blue line is the model control error, yellow line is the EnOI model error, gray line is the GKF model error.

As it is seen from those Figs. the GKF scheme has a substantial advantage comparatively both with EnOI DA scheme and control. Both the forecast and analysis error in GKF method are much smaller, in two-three times less than control and in two times less than EnOI. Also, it is possible to note, that EnOI scheme in generally produces smaller forecast error than control but near day 20 this order is violated. At the same period the forecast error of GKF method always is smaller than control, quasi equaled with the error of EnOI near day 17.  Similar one can say about analysis error. However, it should be pointed out

that both DA method work properly, they both decrease the forecast error and improve the prognostic skill of the model. Also, one may note that the model itself (control) makes the forecast error smaller in time since the model is undergone by the sea surface temperature (SST) nudging (relaxation), taken from the real database NCEP/NCAR. However, this procedure is out of the scope of our paper.

    Figures 2 and 3 show that the major deviation between both of DA methods and control occurs near the day 27, after this

day all curves become practically steady. Therefore, our further analysis concerning model fields and independent data relates to this day, January 27, 2010.

    We start our analysis with the model output of SLA fields.  Figures 4a-d shows the SLA model values, control, both DA analysis and their difference. Figures 5a-d presents the SST also for these schemes.

    All these Figs show the similar structure of SLA in the Atlantic with some specific details. Figure 4a (control) presents the

general dynamic structure with the pronounced positive eddies in Golf Stream zone and slightly expressed negative eddies in Brazilian-Malvinian confluence zone in the Southern Atlantic. Also, it is clearly seen the negative SLA on the equator which also propagates along the South America coastal zone in Caribbean Sea and Guinea Bay.

(a)





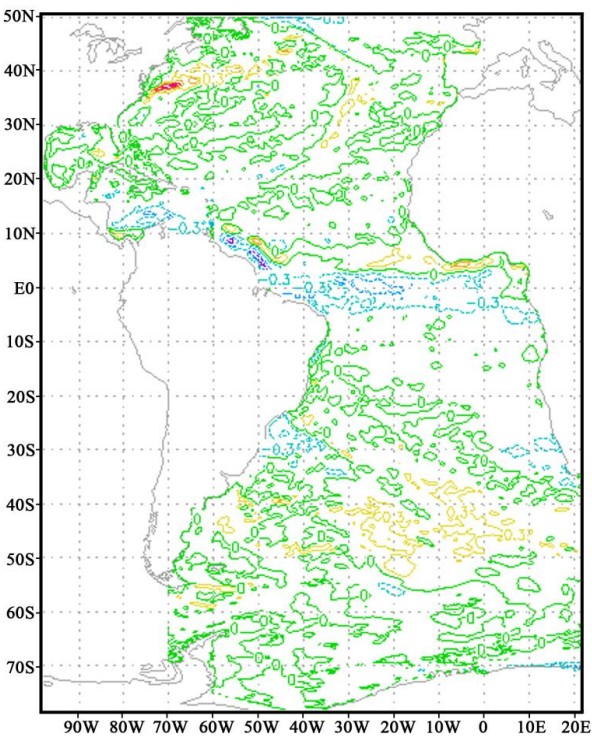

(b)

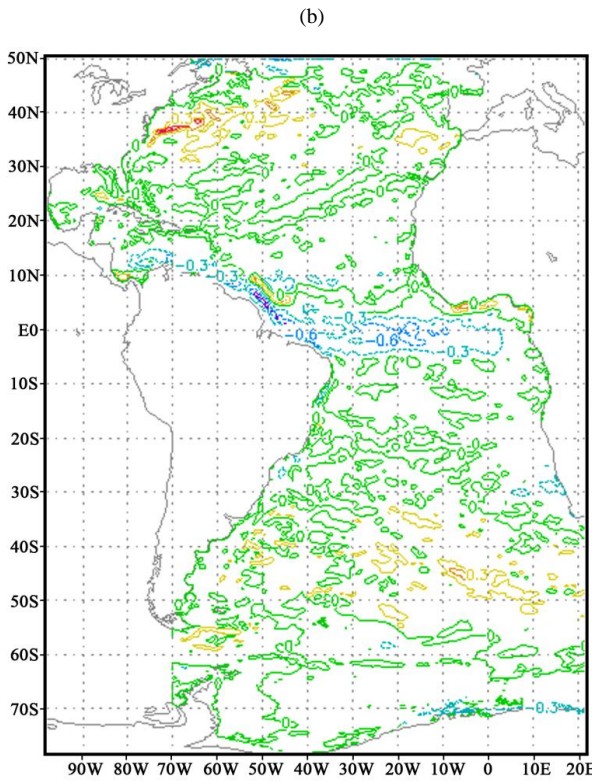

(c)

(d)

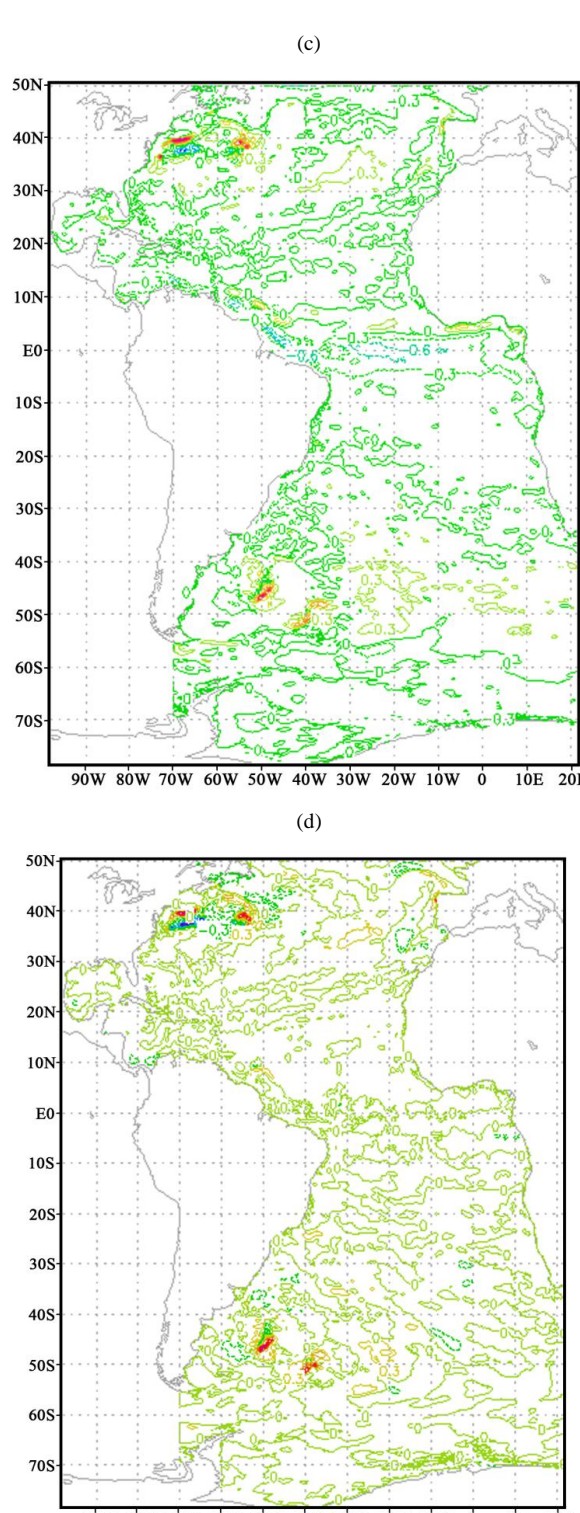

**Figure 4.** SLA after DA computation on 27.01.2010. (a) control, (b) EnOI, (c) GKF,(d) EnOI minus GKF.



EnOI (Kalman) filter DA method (Fig. 4b), in general, repeats the same structure but essentially smooths the positive anomalies above in the Golf Stream Zone and negative anomalies in the Center of Atlantic. This is explainable because the EnOI filter method uses the ensemble statistics on 50 model outputs and they may be locally and instantly quite different.

On a contrary, GKF method substantially intensifies the positive anomalies in the Golf Stream, makes them more compact and, at the same time produces the positive anomaly in the Southern Atlantic near the Magellan passage. This is a temporal instant effect that corresponds to the local trend of SLA in this zone. The difference of two methods (EnOI minus GKF) (Fig. 4d) confirms this conclusion, the positive anomaly does appear neither control, nor EnOI scheme, but presents in GKF method.

    Figure 5 demonstrates the SST structure for the same domain and the same time-moment. As in Fig. 4 it is shown the

control SST (Fig. 5a), SST calculated using the EnOI scheme (Fig. 5b), GKF scheme (Fig. 5c) and their difference (EnOI minus GKF) (Fig. 5d). There is practically no visible difference between the control and EnOI calculations, only few can be noticed in the northern part of Golf Stream zone. On a contrary, there is very essential difference between EnOI and GKF schemes, as it can follow from Fig. 5c and Fig. 5d. This difference is clearly pronounced in the northern part of Atlantic, where warm eddy appears and propagates along the current. This is temporary effect and this meander clearly expressed locally. In

southern Atlantic near the Brazilian-Malvinian confluence zone we also see the strong local dynamics. One also can assert that this is temporary effect which is connected with the instant time variability, that is infinitesimal characteristics of the model vs data.

(a)

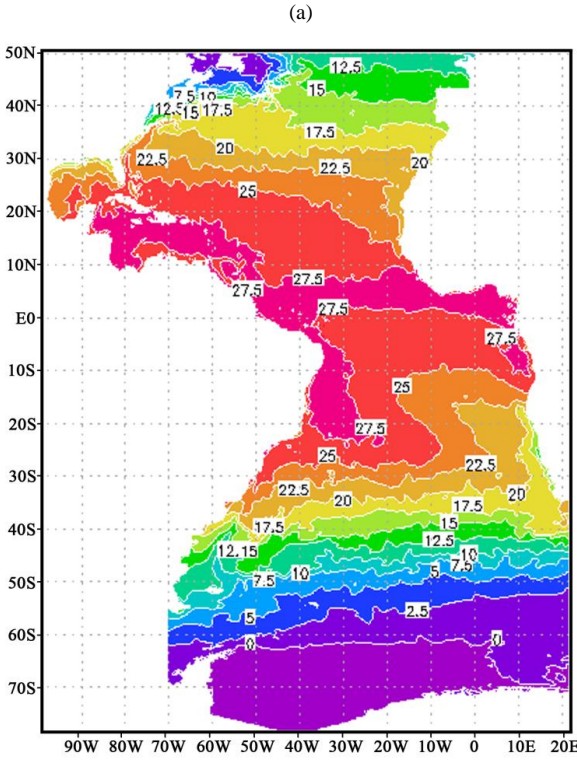


(b)




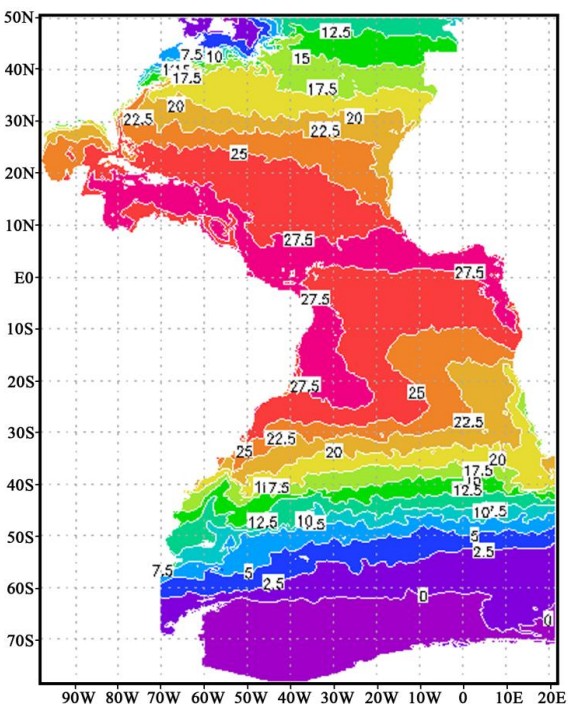

(c)

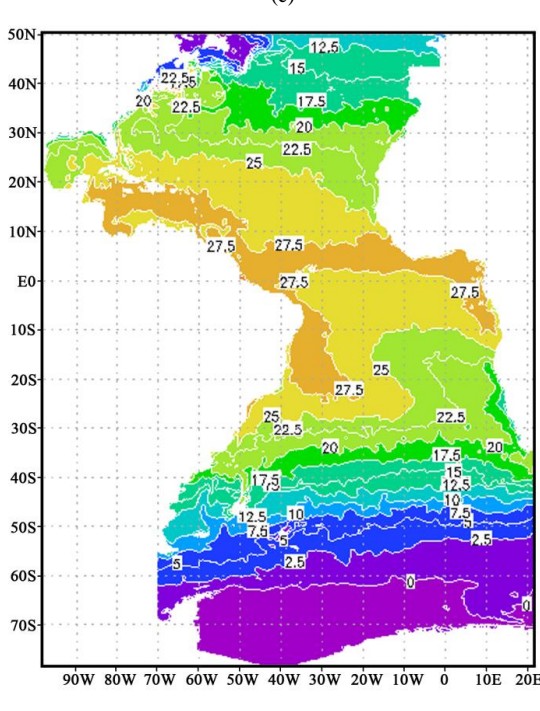

(d)





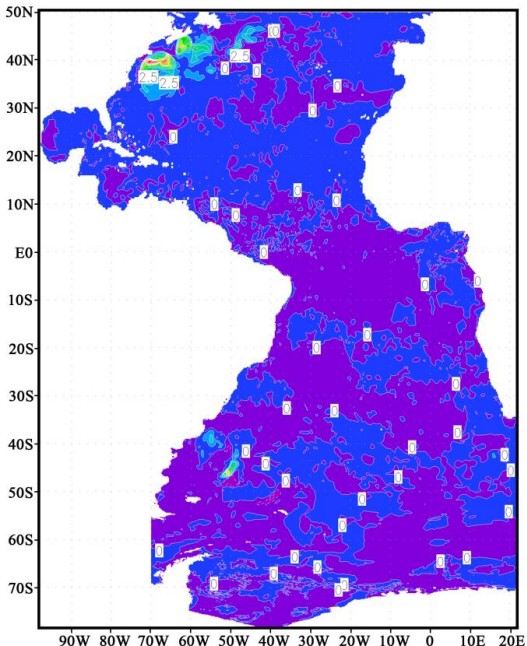

**Figure 5**. SST for all assimilation methods. (a) control, (b) EnOI, (c) GKF, (d) EnOI minus GKF.

**5 Comparison with independent data**

5 **5.1 Comparison with PIRATA moorings**

For comparison we use the data array from PIRATA moorins. As it is well-known there are 17 moored bouys in Tropical and Center Atlantic and data for temperature and salinity from sea surface until 500 m depth are recorded each 15 min and storaged in Internet. Daily data are availiable in website http://pirata.ccst.inpe.br/en/data-2.

The comparison has been realized as follows. On day 27 January the average values from all model results with two DA

10 schemes and control independently have been linearly interoplated onto PIRATA bouys location and level. Then the absolute difference between model and observations totally for all bouys and independently for each level was calculated. These computations have been performed for temperature and salinity separately.

The results of comparison are presented in Fig. 6. Figure 6a contains the deviation for salinity relatively real data and Fig. 6b does the same for temperature. Gray line regards to the control, dashed line shows the deviation of EnOI from reality and

15 blue line corresponds to the deviation of GKF.

(a)

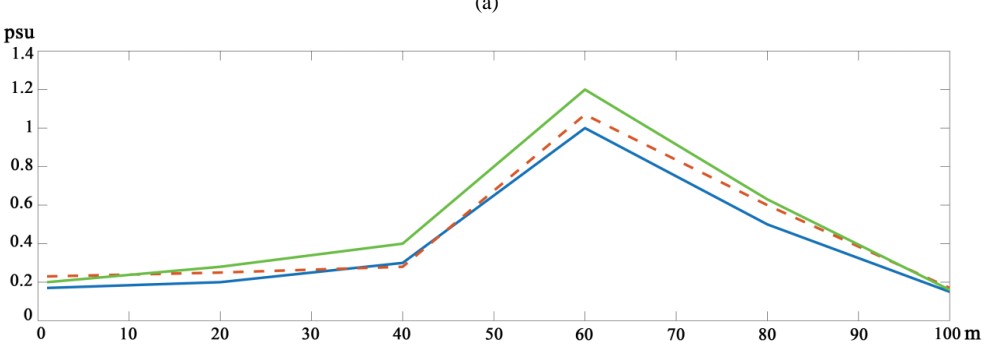

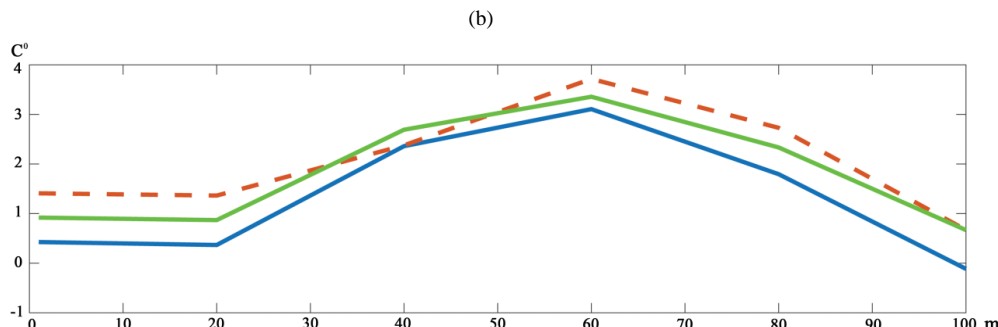

(b)

**Figure 6.** The difference relatively PIRATA moorings.

5    As one can see from these Figs. the GKF method produces the best simulations relatively observations. In Fig. 6a for
salinity the worse result for all methods including control occurs on 40 m depth but for temperature this takes place in 60 m.
Below 100 m temperature simulation is completely wrong and this can be explained due to the model insuficiency to represent
the equatorial under current state. As to salinity the mixed layer is exactly on 40 m and model is unable accurately predict its
position. However, all three scheme including control are similar and reflect reality.

10   **5.2 Comparison with SST observations**

Figure    7    contains    the    SST    map    directly    downloaded    from    webadress    ftp://podaac-
ftp.jpl.nasa.gov/allData/ghrsst/data/L4/GLOB/UKMO/OSTIA/2010/. Then these data were interpolated into model grid. Since
it is difficult to estimate all sources of error we do not show the numerical comparison model and observed SST but simply
present the resulted SST field and make the comparison in quality.

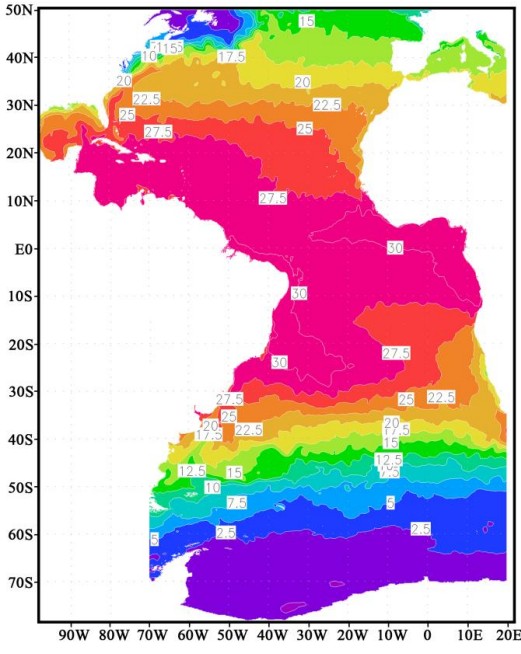

**Figure 7.** Observed SST field.



As it is possible to note the presented SST field contains all feachures characerestic for the Central and South Atlantic and which are shown in Figs. 5a-c. However, Fig. 7 contains also visible sinoptic eddies in the Golf Stream zone as well as in Brazil- Malvinian confluence zone and these eddies are not seen in control and EnOI computations (Figs. 5a-b). On a contrary, Fig. 5c contrains pronounced eddies both in Golf Stream zone and Brazil-Malvinian zone, which match very well to the

observed SST. At least it is possible to assert that GKF method is able to capture the sinoptic variability in the ocean and does it better than EnOI DA scheme and control.

**6 Conclusions and outlook**

The novel and feasible GKF DA method is presented here and it has been compared with the standard EnOI DA scheme which frequently uses in theoretical researchs and practical applications. The study asserts both in quantity and in quality that the

presented GKF scheme has several advantages in comparison with EnOI. In particular, it provides better 24h forecast and better a posteriori analysis. Beside that, the GKF method caputres the ocean sinoptic variability and its dynamics and the ocean temperature and salinity fields calculated after the application of GKF better correspond to independent PIRATA and OSTIA dataset.

*Data availability*. Data sets are available upon request by contacting the correspondence author.

*Author contributions*. The paper was written by KB and AK with the contribution by other authors. Numerical simulation was carried out by IS. Observational data and model integration have been provided and processed by CAST.

*Competing interests*. The authors declare that they have no conflict of interest.

*Acknowegements.* This work was supported by the Russian Science Foundation, project no. 19-11-00076. The Brazilian author separately thanks to Brazilian National Agency of Petroleum, Natural Gas and Biofuels.

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
