# Peer review of "A hybrid data assimilation method and its comparison with an Ensemble Optimal Interpolation scheme in conjunction with the numerical ocean model using altimetry data"

_Ocean Science, 2019_

## Referee Comment (RC2) · Anonymous Referee #2 · 8 Aug 2019

This paper presents a practical implementation of a Kalman Filter (KF) Data Assimilation algorithm variant. The method itself, called afterward in the paper as "GKF" for Generalysed Kalman Filter was previously presented by the same author in a previous paper (Belyaev, K. et al., 2018: An optimal data assimilation method and its application to the numerical simulation of the ocean dynamics). Compared to the most common implementations of the KF, an additional constraint on model and observation temporal tendencies is added when estimating the analysis correction (p.2, l.30).

The manuscript focus on the application of the GKF to a basin scale ocean state estimation. It is intended to illustrate the performance of this "modified" KF compared to the more commonly used Ensemble Optimal Interpolation (EnOI). After a short review of the method basis, results of 1-month experiments assimilating along track SLA observations in an Atlantic Ocean configuration at 1/12° are analyzed. The analyzed fields are compared with the ones obtained from an EnOI approach and an experiment without any data assimilated. Observation misfits, as well as physical fields at a given date are taken as a measure of success for the implemented method.

The manuscript suffers from a poor level of English. It has to be reviewed by a fluent/native English speaker.

It also suffers from a lack of explanations and justifications on the meaning of the hypothesis made on the model and observations, their trend and error to derive the GKF from the KF. The validity of those a priori hypotheses has to be discussed in the context of daily basin scale ocean data assimilation, presented here. This could also help justifying why the GKF is best suited here in theory compared to the EnOI. The way the additional EnOI parameters (alpha and R) are chosen for a fair comparison with the GKF is not discussed and could largely affect the observation misfits.

I would recommend major revisions to improve the manuscript readability but also the justification for the use of the method for daily ocean state estimation and analysis of the results.

Title

I would recommend the use of "Generalized" in the title, instead of "Hybrid" not used again in the text to refer to the proposed scheme.

Abstract

- p.1, l.15: "The method is able ...to produce analysis closer to observations": closer compared to what? "It also conserves the model balance." This property should be explained in the manuscript and not only mentioned here.

- p.1, l.16: "...their errors": the dot at the end of the sentence is missing. The "confidence range of the analysis", mentioned here as an advantage, is neither shown nor discussed later.

The abstract has to be improved to better fit the manuscript content.

Introduction

- p.2, l.5: The constraint on the DA scheme "cost" you mention is mostly relevant in the context of real time production of ocean forecasts.

- p.2, l.10: Some implementations of the 4DVar seek to optimize not only the Initial Conditions but also the boundary conditions (mostly the atmospheric forcing fields).

- p.3, l.4: It the current work...

The assimilation method and the numerical algorithm of its realization

- p.3, l.29: I would suggest following the unified notation introduced by K. Ide et al, 1997 (https://doi.org/10.2151/jmsj1965.75.1B_181), widely used in the atmospheric and oceanographic DA community. The linear dynamical model is then noted M instead of $\Lambda$.

The way you define the observation and model trends has to be better explain and justify.

- P.4, l.10: From my understanding, an observation operator has be applied to x a,n to map the model state into the extended observation space?

Could you explain the use of the ocean model analysis to compute the so called observation trend, since you have access to the observation vector at the analysis step n? And the use of the expectation?

Computational experiments

- p.5, l.18: It is not mentioned in the text that you assimilate "along track" sea level

anomaly and not maps, both types of products being produced by AVISO.

- p.5, l.26: "Below we compare the our GKF assimilation ..."

- p.6, l.4: Do you mean: "An archive with 10 years of completely defined fields... ?" Do those fields are instantaneous or daily mean fields?

- p.6, l25: Could you explain what means $C_{n+1}=0$ and $x \check{\text{I}}_{n+1}= x^a_n$ when applied to daily ocean state data assimilation and to which extend this approximation is valuable or not for the solved problem here? This can also help in understanding why the GKF approach is more appropriate to the type of problem solved here than the EnOI.

Results of the experiments and their analysis

- P.6, l28-29: Reformulate the 1st sentence in a better english. The values you compute are Sea Level Anomalies OR Sea Level "height" (SLA+MDT)? How do you compute the model counterpart to estimate the innovations: at the exact time of the observation from instantaneous model forecast field?

- p.7, figure 1: In the legend of figure 1, could you tell which variable is shown and give its unit?

- P.7, l.9: The term "moment" n is confusing: does it refer to the model time step the closer to the observations OR to the assimilation "step" (in day here)?

- P.7, l.10: "with the total amount of observations equaled to N": the notation N was already used for the number of analysis time step; you cannot used it for the number of observations.

- P.7, l.12-13: SLAf and SLAa values used for the skill computation are instantaneous model counterpart of SLAo values at different model time steps between 0 to 24h when data are available OR do you compare the observations with daily mean model outputs? Does it differ from the way it is done within the assimilation process to compute SLA innovations?

- P.7, Figures 2 and 3: You should add the variable name you show and its unit in the legend. The axis units are also missing.

- P.8, l.12: What is the temporal frequency of the SSH nudging?

- P.8, l.14-15: You should show the full experiment period on figures 2 and 3 to better illustrate/justify your assertion: "the major deviation between both of DA methods and control occurs near the day 27, after this day all curves become practically steady".

- P.8, l.18: For the SST, does the January 27, 2010 coincide with the day with a nudging?

- P.9-10, figure 4: The units in the legend are missing. The values on the isolines are difficult to read, what is the contour interval? A zoom (on the Gulf Stream for example) with the observation positions/values could help to see eddy position/shape differences for the three simulations, compared to the observation values.

- P.11, l.5-8: Could you tell if the SLA features you mention are realistic? Seen in the assimilated observations?

- P.11, l.15: The warm eddies only seen in the GKF analysis are also seen in the SST fields used for the nudging?

- P.12, figure 5: It seems that the colobars differ between the plots c and d. The contour labels in the Gulf Stream area restrict the visibility of the SST fields. A zoom could be useful to highlight smaller/regional differences. The SST difference between the GKF and the EnOI SST analysis (panel d) shows very few "round" shaped patterns with high values. Could you explain that? It looks like there were few very high isolated SLA innovations and you can "see" the signature of the prescribed correlation radius in the one of the analysis correction? Does the SSH field differences has the same kind of round shaped patterns at the same locations?

5 Comparison with independent data

- P.13, l.9: It is unclear how you compute your diagnostic: the model average values are daily mean values of the model fields? Do you have also computed daily mean observations from the instantaneous measurements you downloaded (one measure each 15 min)?

- P.13, l.15: The color of the lines on the figure are green, red and blue, there is no "gray" line as stated in the text. P.14, figure 6: the legend do not mentioned the variables that are shown on the figure and the line corresponding to the different experiments.

- P.14, l.11: This section is very short. As its purpose is to show that the GKF SST analysis is able to capture synoptic variability compared to EnOI analysis, I would suggest moving this comparison to the OSTIA SST where the analysed SST fields are compared (figure 5). A zoom on eddies can be done as the basin map do not allow to see the mentioned eddies (p.15, l.3-4). How does OSTIA compare to the nudged SST fields: the NCEP/NCAR SST do not have such eddies at the same date?

6 Conclusions and outlook

The conclusion is very short and remains very general. It should contain more precise outcomes of the study.

The reference to the "Comparison of Data Assimilation Methods in Hydrodynamics Ocean Circulation Models" by Belyaev K. et al. just published in Mathematical Models and Computer Simulations in July 2019 could be added to the list of reference. I found the method presentation clearer in this previous paper.

---

## Author Comment (AC1) · 21 Aug 2019

andrew\_kuleshov@mail.ru

Received and published: 21 August 2019

Authors response to interactive comment Referee #1 on "A hybrid data assimilation method and its comparison with an Ensemble Optimal Interpolation scheme in conjunction with the numerical ocean model using altimetry data" by Konstantin Belyaev et al.

The manuscript presents an application of a recently formulated method, the GeneralÂňized Kalman Filter (GKF), to the assimilation of altimeter data into an eddy permitting model of the Atlantic Ocean. The results in terms of analysis and forecast error variÂňance are compared against a simulation and an assimilation based on an Ensemble Optimal Interpolation method (EnOI). When evaluated with the assimilated altimeter data, GKF is shown to perform better for both metrics than the EnOI. The comparison is kept to the minimum and only the evolution of the globally averaged metrics and fields of one time instance are shown. The additionally impact on SST provides very little inÂň formation except that GKF is able to produce larger changes in SST on eddy-scales than EnOI. Whether they go into the right direction is not possible to judge. Overall the amount of verification would be appropriate for an illustration in a paper that presents the method, which however has already been published. The presentation is very poor particularly due to the abundance of grammatical errors. I stopped to list the necesÂňsary changes on page two, since I felt that the manuscript would require too much extra work. The method remains obscure particularly because of deficiencies in the presenÂňtation. There are also problems in the application (potentially also in the formulation) of the method noted. Notably the application of the Birkhoff-Khinchin theorem on the drifts does not make sense to me as substantiated below. Additionally, the application of the EnOI seems to be problematic since it depends on two parameters whose effects have not been explored and their choices have neither been stated nor the selection criteria explained. One parameter, the scaling factor for the error covariance, seems to be incorrectly implemented, the other parameter, the error covariance of the obserÂňvations, does not appear to be relevant in GKF, which hardly can be correct unless it is assumed to be zero there. The application of EnOI actually demonstrates that EnOI as almost a failure, not being able to adjust the SSS on eddy-scales. This is surprising since this method has been proven to work well with this model before in the Gulf of Mexico. I would expect that the analysis error could be almost arbitrarily reduced with decreasing observational error R, which means that the performance of GKF relative to EnOI can be adjusted by varying R in a way that desired result is obtained. This could be for instance be the one shown in the manuscript: EnOI reduces the analzsis error but less than GKF. Last, it remains

unclear in what way GKF is a hybrid method, since this term is only mentioned in the Title and the Abstract and no further motivation or explanation are given.

Re: The Authors are grateful to Referee for the useful comments, which helped us to improve substantially the paper. The paper was substantially revised to improve its style and grammar. Concerning the specific remarks, we can answer the following: 1. About the application of the Birgkoff-Khinchin theorem. We specially note that in our case when we consider quite a large domain, this theorem is applicable if the convergence is considered in an integral metric, for instance in L2 metric. 2. About the EnOI parameters  $\alpha$  and R. In our paper we have specially noted in conclusions that the GKF scheme is governed by the same parameters. The problem of defining the best parameters for EnOI to adjust the data was not the goal of our study. However, both methods can be compared with respect to these parameters separately. 3. Concerning the failure of EnOI we have showen the opposite, the EnOI works and works correctly, as well as GKF. However, it was shown that the GKF has advanteges over the EnOI. May be it can be done working better, but it has not been our goal. Our goal was to compare both methods and it seems we did this correctly mathematically and methodologically. 4. Why GKF is a hybrid method. When we introduced GKF we have showen that GKF minimizes the given functional and therefore it can be considered as a variational method. At the same time, it uses the statistical approach since it deals the probability theory and methodology and thus it can be attributed to the stochastic methods (Belyaev et al., 2018).

P1 L14-15: How is the ability to assimilate data being judged? I guess by producing a result that this closer to the observation after assimilation. I suggest to connect the two parts into one and make clear relative to what state the analysis is "closer" L20: remove "a" from "A data assimilation" and "a great" L22 "the" in front of "Australian" and "American" L24 "altogether" instead of "at all" L25 remove "on how" L31 "Ghil et al (1991)" L34 "led" L35 remove "the" before "data L36 replace "the" before essential by "an" Re: We agree with all remarks and the text was revised according to them.

СЗ

The changes are marked in red. P2 L1 replace "indicate" with "name" L2 add "the" in front of "Australian" and "American" L5 "economic" instead of "economy" L6 "medium" instead of "media" L7 add "the" before "used" L9 add "a" before "given" and change "with respect to" to "represented by" L12 replace "seeking" with "sought" and replace "in a" by "depending on" L13 research is uncountable and cannot be used with several replace "be found in (Marchuk" with " be found (e.g. Marchuk" L16 not clear what the sought field is in this context. It was the initial state in the previous paragraph but minimizing is variance would not make sense to me. Do you mean variance or variation, it the latter case it seems to be just the variational method. L17 I think it should be "the observed variables are the sum of the true signal, which the model is supposed to represent, and stochastic ...." L18 remove "the" before "known" L19 should be "Pendruff et al. (2002)" this is one of the modern development but not the only one. L20 sloppy formulation, should read "to the papers of Evensen (2009) and Xie et al (2010)" L22 I don't understand what that means. Change to "Van Leeuwen (2015)" and "Van Leeuwen (2011)", respectively L25 remove "the" before "hybrid and "both" and replace "are" with "have" L26 add "a" before "functional" L28 should be " found (e.g. Lorenc et. al., 2015; Tanajura et. al., 2013). We may also refer to Tanajura et. al. (2009)," L34 In the EnKF you have and ensemble of model runs but only one representation of the observations. Ensembles of observations are the same as just observations. L35 "As a consequence..." is this a problem or the goal of the method? In the latter case it should be formulated differently. I36 Not clear why it becomes better if no assimilation occurs. Better in the sense of what? Numerical forecast does not make sense to me because this might be ultimately happening but I don't see how this as a direct consequence. It could be better in the sense of obeying the dynamical equations L37-40 Minimum energy is characteristic for equilibrium states. For transitions I would apply conservation principles, e.g. those of mass or momentum. This would be equivalent to minimum energy change. Maybe this is meant here? Re: We agree with all remarks and the text was revised according to them. P3 L4-5 Please motivate why this method is used for comparison and not the EnKF as a more advanced method. Is this is more similar to the GKF? Re: We used EnOI for a comparison with GKF because we used the same logic to prepare the initial ensemble(s). Both EnOI and GKF use the previously prepared ensemble statistics to create the covariances but in case of GKF we use two ensembles in two sequential time-moment and EnOI uses one. In a case of EnKF we would need to create independent ensembles for each time moment, but for GKF we could do the same in two sequential time moments. L7 "data" instead of archive Re: We agree and the text was revised. L8 Strange formulation, the following is not about the possibility. Re: We agree and the text was revised.

I14 I don't get where the Fokker-Planck equation may play a role in this comparison. Eq. (3) eq (3) doesn't make sense from the dimensional point of view. The first () is a scalar, the second () is a vector of the model-dimension while the third () is a vector of observational dimension and Q a square matrix of observational dimensions. Re: Reference to the Fokker-Planck equation has been removed. Formula (3) is correct:  $\sigma$  is a scalar;  $\Lambda$  and C are both vectors (column), dim r×1, where r is the model dimension; is a vector string, dim 1×m, where m is dimension of observations; Q is a matrix, dim m×m. Totally we have K is matrix with dimension r×m.

P4 L6 I suggest to use "the model variables that are observed". "the observed variables" are understood as the observations. Re: We agree and the text was revised. I9 As the observational trend I would expect the difference Yn+1 - Yn rather than the difference between model and data Re: We agreed with this comment and rephrased the sentences. L10-11 should read "coincides". The operator for the "prolongation" needs to be speciÂňfied. Prolongation is as term that refers to time, while I have the impression that actually interpolation/extrapolation in (phase) space is more relevant here. Re: We agreed with this comment and rephrased the sentences. L11-12 I don't understand the ensemble average. This requires that something is obÂňserved in multiple ways which is rarely possible, except for if two satellite tracks cross. I think it should be made clear from the beginning that your ensemble is generated from several time instances before the assimilation step Re: We agreed and changed the text.

L13 this Y needs a different symbol than the one above on line 10 because they have different dimensions. Also Yn+1 depends on Cn+1 while Cn+1 depends on Yn+1, so it is not clear how this could work Eq.5 Why is the formula for Cn+1 different here from line 10 ? Re: In this context Y is the observations, is the extrapolated observational vector on the entire space, is the anomaly of observed vector relative to vector C projected onto observational space. Actually, observations are divided into two parts: one part is related to trend C and the resudial part is related to innovation and assimilates. This distinguishes GKF method from EnOI where there is no trend and observations are assimilated as a whole. L32-33 "In the expanded form": If different forms for the equations exist this should be made clear from the beginning and transition operators defined. Re: In Eq. (6) the vector-column is multiplied by the vector-string and it defines the algorithm of calculation of the matrix Q. L37 you mean Eq. (5) Re: We agreed and changed the text. I36-37 I don't think this theorem applies for the cases that you have in mind. If the system is stationary there should be no trend. Eq. (5) provides an estimate of the tendency at a particular time. This is supposed to change with time. While Eq.(7) will change less and less with progressing time. For a constant tendency (that means slowly varying changes) (7) will even have a different sign because most of the Xai will soon be in the past relative to n. Re: This comment is correct if we consider the variability at each point independently. However, in our case we consider the state vector X on quite a large enough domain. Therefore, the positive trend at one point is compensated by the negative trend at another point (this is a reasonable suggestion). This means that the Birgkoff-Khinchin theorem is valid place if the convergence is considered in L2 metric (quadratic mean). P5 L9 The layers should have different but uniform densities. Re: We agreed and changed the text. L17 innovative use of language, but for normal people better remove relatively and just say "the temporal average ... has been removed" Re: We agreed and changed the text. L19 Previously ... " If you did this you have to describe what has been done otherwise a suitable reference should be provided. Re: We used the data from AVISO and the phrase that the data have passed the quality control was removed from the text. L22-23 I don't understand

what this has to do with parallelization. After parallelization you should be able to handle larger problems. And what does "reduced" mean - what has been done to reduce the size. Re: We agreed and changed the text. L25 why not say what they use? Re: We agreed and changed the text. L26-27 Particularly here you need to provide details of the scheme that was used to create the data for comparison. It is not even clear whether the respective results are discussed by Tanajura et al (2015) or elsewhere. Re: All details on how we provide data for comparison are described in 3.3. L33-34 "The high parallelization..." I don't understand what this means? Re: We agreed and changed the text.

P6 L3-4 no freshwater fluxes? Is there at least surface SSS relaxation? Re: We did not mention the freshwater flux conditions because it does not play any role in our research. Here we can say that the freshwater flux forms from rivers and precipitations. Rivers are climatological with annual cycle precipitations given from the reanalysis. Sea surface salinity has a relaxations which are formed from climatological data (ATLAS) and model computations.

L6 what does real mean. Are the without error. Re: real wind stress was taken from GFS without any changes.

L14 I don't understand how you get the number 50, do I need to know the time step for this? Re: Our previous results have shown that sample of 50 members is sufficient for statistically reasonable conclusions. It would be computationally expensive and unreasonable in this context, to extend substantially the sample's length. Eq.(8) in the ensemble OI the factor alpha also appears inside the () in front of H thereby acknowledging the fact that the variance over time typically is too large. L24 what are these considerations and how are the actual values for R and alpha? Why is there no error covariance matrix R necessary in GKF? Does it mean GKF uses zero error? Re: The factor alpha as well as matrix R are two empirical parameters for EnOI. The matrix Q in our method plays the same role as the matrix (HBHT + R) in EnOI. The only difference is that our scheme the matrix Q contains the anomaly relative to the previous model

state (already constructed analysis), while EnOI considers the anomalies relative to the average model state (matrix B). Therefore, the matrix Q can also be considered as Q=Q1+R, where R an empirically given variance. However, in our case, this pure empirical parameter can be compensated by the normalized factor sigma, (Eq. (3)) while in EnOI this is impossible. The same can be said for parameter alpha. The discussion about this issue has been already presented in Belyaev et al. (2018). In the EnOI scheme  $\alpha$  was chosen 1 and R is diagonal matrix with values 0.01. All conditions which are necessary to apply GKF are discussed before (see Re in P4).

P7 Figure 1 which day is this and which model run? Re: Fig. 1 shows only an example the information of which we use. The HYCOM model and AVISO data set are shown together. The day when it was calculated is not specified. L15 better: initialized from the analysis at the past time n-1 Re: We agreed and changed the text. L16 simply: SLAa is the analysis at the time n Figure 2/3 what are the units on both axes? Are these days, which means the forecast period is one day? The forecast period should explicitly stated somewhere Re: We agreed and changed Figures 2,3.

P8 L5 How do the results depend on the heuristically defined values alpha and R in the EnOI? Can EnOI get closer to the data with a larger alpha and/or a smaller R, particuÂňlarly since R seems to be zero in GKF? Re: We did not specifically investigate how the result of EnOI application depends on  $\alpha$  and R, because this is another problem, Because we used the same parameters in both EnOI and GKF. L14-15 My interpretation of Figs.2/3 is that the curves are displayed only until day 27 therefore I can not verify this based on Figure 2 and 3 Re: We agreed and changed Figures 2,3. Figure 4:please provide units and a colorbar the I can only read 0.3. Re: We agreed and changed Figures 4. P 11 L4-5 What is the argument here, the states are too different therefore B too large? Maybe the reason is the omission of alpha inside of the brackets of Eq. () or an inapÂňpropriate value for alpha. EnOI hardy improves the state in regions where the variability is large. Re: We did not state that B is too large. On the contrary we show that the EnOI method works in a right direction and really assimilates data and decreases the error. In addition, We show that the GKF method does the same and a little bit better. We already said that the alpha and R parameters are the same in both methods. P13 I10 Why not the RMS differences instead of the mean bias, the mean bias across all moorings is not so interesting. Re: We used the mean bias but if we use the RMS metric there would be no difference in quality, only in quantity. But the mean bias demonstrate the differences between the methods more clearly. I 14-15 move or copy to the captions. Re: We agreed and changed caption to Figures 6.

P14 I1-6 This comparison is too superficial to be helpful since SST is only shown to demonÂňstrates only the existence of corrections on eddy-scales. Re: Fig for SST shows that GKF really corrects the model SST and makes it more dynamic. This confirms our previous results and those obtained by applying other DA schemes and does not contradict to observations. I9-10 I would have hoped that the study provides evidence rather than just asserting something. Re: We tried to prove our results not only mathematically and theoretically but also performing numerical experiments and comparing obtained results with observations.

Please also note the supplement to this comment: https://www.ocean-sci-discuss.net/os-2019-56/os-2019-56-AC1-supplement.pdf

---

## Author Comment (AC2) · 21 Aug 2019

Authors response to interactive comment Referee #2 on "A hybrid data assimilation method and its comparison with an Ensemble Optimal Interpolation scheme in conjunction with the numerical ocean model using altimetry data" by Konstantin Belyaev et al.

This paper presents a practical implementation of a Kalman Filter (KF) Data Assimilation algorithm variant. The method itself, called afterward in the paper as "GKF" for

[Figure]

Generalysed Kalman Filter was previously presented by the same author in a previous paper (Belyaev, K. et al., 2018: An optimal data assimilation method and its application to the numerical simulation of the ocean dynamics). Compared to the most common implementations of the KF, an additional constraint on model and observation temporal tendencies is added when estimating the analysis correction (p.2, l.30). The manuscript focus on the application of the GKF to a basin scale ocean state estimation. It is intended to illustrate the performance of this "modified" KF compared to the more commonly used Ensemble Optimal Interpolation (EnOI). After a short review of the method basis, results of 1-month experiments assimilating along track SLA observations in an Atlantic Ocean configuration at 1/12_ are analyzed. The analyzed fields are compared with the ones obtained from an EnOI approach and an experiment without any data assimilated. Observation misfits, as well as physical fields at a given date are taken as a measure of success for the implemented method. The manuscript suffers from a poor level of English. It has to be reviewed by a fluent/ native English speaker. It also suffers from a lack of explanations and justifications on the meaning of the hypothesis made on the model and observations, their trend and error to derive the GKF from the KF. The validity of those a priori hypotheses has to be discussed in the context of daily basin scale ocean data assimilation, presented here. This could also help justifying why the GKF is best suited here in theory compared to the EnOI. The way the additional EnOI parameters (alpha and R) are chosen for a fair comparison with the GKF is not discussed and could largely affect the observation misfits. I would recommend major revisions to improve the manuscript readability but also the justification for the use of the method for daily ocean state estimation and analysis of the results.

Re: The paper was substantially revised to improve its style and grammar.

About the EnOI parameters $\alpha$ and R. In our paper we have specially noted in conclusions that the GKF scheme is governed by the same parameters.

Title I would recommend the use of "Generalized" in the title, instead of "Hybrid" not
used again in the text to refer to the proposed scheme. Re: We agree with the remark and the title is changed. All changes are marked in blue.

Abstract - p.1, l.15: "The method is able ...to produce analysis closer to observations": closer compared to what? "It also conserves the model balance." This property should be explained in the manuscript and not only mentioned here. Re: We agree and the text was revised. - p.1, l.16: "...their errors": the dot at the end of the sentence is missing. The "confidence range of the analysis", mentioned here as an advantage, is neither shown nor discussed later. Re: The explanation about confidence range has been added in the Conclusion. The abstract has to be improved to better fit the manuscript content. Re: We changed the abstract and it seems to fit better to the paper content. Introduction - p.2, l.5: The constraint on the DA scheme "cost" you mention is mostly relevant in the context of real time production of ocean forecasts. - p.2, l.10: Some implementations of the 4DVar seek to optimize not only the Initial Conditions but also the boundary conditions (mostly the atmospheric forcing fields). Re: We agree and the text was revised. - p.3, l.4: It the current work... Re: This misprint is fixed.

The assimilation method and the numerical algorithm of its realization - p.3, l.29: I would suggest following the unified notation introduced by K. Ide et al, 1997 (https://doi.org/10.2151/jmsj1965.75.1B_181), widely used in the atmospheric and oceanographic DA community. The linear dynamical model is then noted M instead of _. Re: In our case we would not like to change the notations which we already used before in several published papers. Besides that, standard notations M is used for the model forecast (background), while in our case, Lambda is used for the time-derivative (infinitesimal part). The way you define the observation and model trends has to be better explain and justify. Re: This remark has been already accounted when we replied to referee 1 and we already did several amendments in the text. - P.4, l.10: From my understanding, an observation operator has be applied to x a,n to map the model state into the extended observation space? Re: The operator H (projection) really applied to the analysis X (a,n) and to vector C (n+1) which includes the analysis X(a,n). The

extrapolation is applied to the observational vector and extends the observations to the entire model space. We have done this by adding the model ensemble statistics which complements the unobserved variables. Could you explain the use of the ocean model analysis to compute the so called observation trend, since you have access to the observation vector at the analysis step n? And the use of the expectation? Re: Indeed, we use mathematical expectation (model ensemble average statistics) to complement the unobserved variables. This allows calculating the observational trend as the difference between the new constructed vector as we described before and previously calculated model analysis.

Computational experiments - p.5, l.18: It is not mentioned in the text that you assimilate "along track" sea level anomaly and not maps, both types of products being produced by AVISO. Re: We agree and the text was revised. - p.5, l.26: "Below we compare the our GKF assimilation ..." Re: This remark has been already accounted when we replied to referee 1 and we already did several amendments in the text. - p.6, l.4: Do you mean: "An archive with 10 years of completely defined fields... ?" Do those fields are instantaneous or daily mean fields? Re: Data AVISO are daily mean values. - p.6, l25: Could you explain what means $C_{n+1}=0$ and $x$ ÌËĞE $_{n+1}= x_a$ $_n$ when applied to daily ocean state data assimilation and to which extend this approximation is valuable or not for the solved problem here? This can also help in understanding why the GKF approach is more appropriate to the type of problem solved here than the EnOI.

Re: We explained earlier that $C(n+1) =0$ means that the anomalies are taken with respect to the model average value(s) and there are no trends (model and observations) considered. This is the main difference between EnOI and GKF. This explains why in daily assimilation GKF is more powerful then EnOI, since it accounts not only the average state but also the daily dynamics.

Results of the experiments and their analysis - P.6, l28-29: Reformulate the 1st sentence in a better english. Re: We agree and the text was revised. The values you compute are Sea Level Anomalies OR Sea Level "height" (SLA+MDT)? How do you

compute the model counterpart to estimate the innovations: at the exact time of the observation from instantaneous model forecast field? Re: We removed model average to obtain the anomalies and calculate the innovation subtracting the SLA and model values without average projected onto observational point on track. We explained in text that the special procedure of bias correction has been applied and described this procedure in detail. - p.7, figure 1: In the legend of figure 1, could you tell which variable is shown and give its unit? Re: We agree and the legend of figure 1 was revised.

- P.7, l.9: The term "moment" n is confusing: does it refer to the model time step the closer to the observations OR to the assimilation "step" (in day here)? Re: We agree and the text was revised. - P.7, l.10: "with the total amount of observations equaled to N": the notation N was already used for the number of analysis time step; you cannot used it for the number of observations. Re: We agree and replace N by L. - P.7, l.12-13: SLAf and SLAa values used for the skill computation are instantaneous model counterpart of SLAo values at different model time steps between 0 to 24h when data are available OR do you compare the observations with daily mean model outputs? Does it differ from the way it is done within the assimilation process to compute SLA innovations? Re: We compare the SLA model output after assimilation using EnOI, GKF and free run at each observational points; f-means 24h forecast, a-means after assimilation at the same assimilation moment. - P.7, Figures 2 and 3: You should add the variable name you show and its unit in the legend. The axis units are also missing. Re: We agree and the legend and Figures 2,3 was revised. - P.8, l.12: What is the temporal frequency of the SSH nudging? Re: SSH nudging is not used for this version of HYCOM. - P.8, l.14-15: You should show the full experiment period on figures 2 and 3 to better illustrate/justify your assertion: "the major deviation between both of DA methods and control occurs near the day 27, after this day all curves become practically steady". Re: We agree and the Figures 2,3 were revised.

- P.8, l.18: For the SST, does the January 27, 2010 coincide with the day with a nudging? Re: Nudging for SST is performed during all period of computations. - P.9-10,

figure 4: The units in the legend are missing. The values on the isolines are difficult to read, what is the contour interval? A zoom (on the Gulf Stream for example) with the observation positions/values could help to see eddy position/shape differences for the three simulations, compared to the observation values. Re: We agree and the Figure 4 was revised. - P.11, l.5-8: Could you tell if the SLA features you mention are realistic? Seen in the assimilated observations? Re: The SLA obtained after assimilation showed the intensive dynamics, especially in the Golf Stream zone and intensification of synoptic variability. They are realistic and confirmed not only in our study. - P.11, l.15: The warm eddies only seen in the GKF analysis are also seen in the SST fields used for the nudging? Re: In these eddies also are seen but not so intensive as in Fig. 4. - P.12, figure 5: It seems that the colobars differ between the plots c and d. The contour labels in the Gulf Stream area restrict the visibility of the SST fields. A zoom could be useful to highlight smaller/regional differences. The SST difference between the GKF and the EnOI SST analysis (panel d) shows very few "round" shaped patterns with high values. Could you explain that? It looks like there were few very high isolated SLA innovations and you can "see" the signature of the prescribed correlation radius in the one of the analysis correction? Does the SSH field differences has the same kind of round shaped patterns at the same locations?

Re: Fig. 5d is correct. The round "spots" are simply the digits showing the values of contours. In the text there were the following description. This difference is clearly pronounced in the northern part of Atlantic, where warm eddy appears and propagates along the current. This is a temporary effect and this meander is clearly expressed locally. In the southern Atlantic near the Brazilian-Malvinian confluence zone we also see the strong local dynamics. One also can assert that this is a temporary effect which is associated with the instant time variability, that is infinitesimal characteristics of the model vs data. Zooming of this Figure 5d clearly shows these results. Zooming of the Fig 5d clearly shows what we are talking about.

5 Comparison with independent data - P.13, l.9: It is unclear how you compute your

diagnostic: the model average values are daily mean values of the model fields? Do you have also computed daily mean observations from the instantaneous measurements you downloaded (one measure each 15 min)? Re: We computed daily mean values for both model and observational data. - P.13, l.15: The color of the lines on the figure are green, red and blue, there is no "gray" line as stated in the text. P.14, figure 6: the legend do not mentioned the variables that are shown on the figure and the line corresponding to the different experiments. Re: We agree and replace "gray" by "green". - P.14, l.11: This section is very short. As its purpose is to show that the GKF SST analysis is able to capture synoptic variability compared to EnOI analysis, I would suggest moving this comparison to the OSTIA SST where the analysed SST fields are compared (figure 5). A zoom on eddies can be done as the basin map do not allow to see the mentioned eddies (p.15, l.3-4). How does OSTIA compare to the nudged SST fields: the NCEP/NCAR SST do not have such eddies at the same date? Re: We explained that the GKF better captures the synoptic variability than EnOI in Fig. 5. In Fig. 6 we simply showed that this conclusion does not contradict to the observational SST fields.

6 Conclusions and outlook The conclusion is very short and remains very general. It should contain more precise outcomes of the study. The reference to the "Comparison of Data Assimilation Methods in Hydrodynamics Ocean Circulation Models" by Belyaev K. et al. just published in Mathematical Models and Computer Simulations in July 2019 could be added to the list of reference. I found the method presentation clearer in this previous paper.

Re: We extended the section and added a reference to the recently published paper Belyaev et al. (2019) (see P.2).

The authors are grateful to the Referee for useful comments, which helped us to improve substantially the paper.

Please also note the supplement to this comment:

https://www.ocean-sci-discuss.net/os-2019-56/os-2019-56-AC2-supplement.pdf

[Figure]

[Figure]

**Fig. 1.** Figure 5. SST for all assimilation methods. (a) control, (b) EnOI, (c) GKF, (d) EnOI minus GKF.